# Image Stitching Based on Color Difference and KAZE with a Fast Guided Filter

**DOI:** 10.3390/s23104583

**Published:** 2023-05-09

**Authors:** Chong Zhang, Dejiang Wang, He Sun

**Affiliations:** Changchun Institute of Optics, Fine Mechanics and Physics, Chinese Academy of Sciences, Changchun 130033, China; chongzhang17@mails.jlu.edu.cn (C.Z.);

**Keywords:** image stitching, KAZE, RANSAC, fast guided filter

## Abstract

Image stitching is of great importance for multiple fields such as moving object detection and tracking, ground reconnaissance and augmented reality. To ameliorate the stitching effect and alleviate the mismatch rate, an effective image stitching algorithm based on color difference and an improved KAZE with a fast guided filter is proposed. Firstly, the fast guided filter is introduced to reduce the mismatch rate before feature matching. Secondly, the KAZE algorithm based on improved random sample consensus is used for feature matching. Then, the color difference and brightness difference of the overlapping area are calculated to make an overall adjustment to the original images so as to improve the nonuniformity of the splicing result. Finally, the warped images with color difference compensation are fused to obtain the stitched image. The proposed method is evaluated by both visual effect mapping and quantitative values. In addition, the proposed algorithm is compared with other current popular stitching algorithms. The results show that the proposed algorithm is superior to other algorithms in terms of the quantity of feature point pairs, the matching accuracy, the root mean square error and the mean absolute error.

## 1. Introduction

Currently, image stitching is still a topical research issue and is widely used in computer vision, unmanned aerial vehicle (UAV) reconnaissance and other fields [1,2]. Image stitching consists of synthesizing different types of images captured by different imaging devices at different shooting positions into an image with a larger field of vision [3,4]. On the one hand, image stitching can synthesize panoramic and ultra-wide-view images, so that ordinary cameras can achieve grand scene shooting. On the other hand, image stitching can synthesize fragmented images into a complete image. The stitching technology can be applied to combine medical images, scientific microscope fragments or local images from seabed exploration into a complete image. In addition, image mosaic is also a basic technology in scene rendering methods, which uses panoramic images instead of 3D scenes to model and draw. However, image stitching still encounters many challenges at present, such as parallax caused by viewpoint change, ghosting, distortion, detail distortion and image unevenness [5,6,7].

Up to now, many image stitching methods have been proposed by researchers. Li et al. [8] proposed a model of appearance and motion variation based on the traditional speeded up robust features (SURF) algorithm, which mainly contains Hessian matrix construction, Hessian matrix determinant approximation calculations and non-maximal suppression determination feature points in its key steps. In order to improve the matching accuracy and robustness, Liu et al. [9] introduced an improved random sample consensus (RANSAC) feature image matching method based on SURF. First of all, the SURF method is used to detect and extract image features, and the fast library of the approximate nearest neighbor-based matcher method is used to perform initial matching on image feature points. The RANSAC algorithm is improved to increase the probability of correct matching points. In 2017, Guan et al. [10] presented an interest point detector and binary feature descriptor for spherical images. Inspired by the Binary Robust Invariant Scalable Keypoints (BRISKs), they adapted the method to operate on spherical images. All of the processing is intrinsic to the sphere and avoids the distortion inherent in storing and indexing spherical images in a 2D representation. Liu et al. [11] developed the BRISK_D algorithm, which effectively combines features from the Accelerated Segment Test (FAST) and BRISK methods. The keypoints are detected by the FAST algorithm and the location of the keypoint is refined in scale and space. The scale factor of the keypoint is directly computed with the depth information of the image. Zhang et al. [12] proposed a screening method based on binary mutual information for the mismatch problem. The feature points extracted by the ORB algorithm are distributed in a color change area. Then, the new feature points are obtained by internal points. In this way, feature points can be eliminated and the best transformation matrix can be obtained by using an iterative method. In 2020, Ordóñez et al. [13] introduced a registration method for hyperspectral remote sensing images based on MSER, which effectively utilized the information contained in different spectral bands. Elgamal et al. [14] proposed an improved Harris hawks optimization by utilizing elite opposite-based learning and proposing a new search mechanism. This method can avoid falling into local optimum, improve the calculation accuracy and accelerate the convergence speed. Debnath et al. [15] utilized Min Eigen feature extraction based on the Shi-Tomasi corner detection method which detects interest points to identify image forgery.

With the continuous development of machine learning, many scholars have applied it to the field of image stitching. However, learning-based image stitching solutions are rarely studied due to the lack of labeled data, making the supervised methods unreliable. To address this limitation, Lang Nie et al. [16] proposed an unsupervised deep image stitching framework consisting of two stages: unsupervised coarse image alignment and unsupervised image reconstruction. The stitching effect of this method is more obvious for images with few features or low resolution. However, stitching algorithms based on machine learning require extensive training, have a high resource consumption and are highly time consuming. 

Alcantarilla et al. [17] introduced the KAZE feature in 2012. Their results revealed that it performs better than other feature-based detection methods in terms of detection and description. Pourfard et al. [18] proposed an improved version of the KAZE algorithm with the accelerated robust feature (SURF) descriptor for SAR image registration. Discrete random second-order nonlinear partial differential equations (PDEs) are used to model the edge structure of SAR images. The KAZE algorithm uses nonlinear diffusion filtering to build up the scale levels of the SIFT descriptor. It preserves edges while smoothing the image and reduces speckle noise.

For the purpose of alleviating the sensitivity of KAZE and improving the stitching effect, we proposed image stitching based on color difference and KAZE with a fast guided filter as an efficient stitching method. The contributions of this paper are as follows: firstly, we introduce a fast guided filter into the KAZE algorithm to effectively reduce the mismatch information and improve the matching efficiency. Secondly, we use the improved RANSAC algorithm to increase the probability of correct matching points sampled and effectively eliminate the wrong matching point pairs. Thirdly, we introduce color difference and brightness difference to compensate for the whole image when fusing and stitching images. This method cannot just only effectively eliminate seams, but also results in perfectly uniform color and luminance.

The remainder of the paper is organized as follows. Section 2 depicts the fast guided filter and Additive Operator Splitting (AOS) algorithm briefly. Section 3 details the proposed method in this paper. Section 4 presents the experimental results and assessments. Finally, the conclusions and outstanding issues are listed in Section 5.

## 2. Related Work

In this section, we introduce the fast guided filter, nonlinear diffusion and the AOS algorithm briefly.

### 2.1. Fast Guided Filter

A guided filter is an adaptive weight filter which can restore the boundaries while smoothing images via a guide map [19]. Assume that the input image, the output image and the guidance map are expressed as p, q and I, respectively. A guided filter driven by a local linear model can be defined as shown in Equation (1):(1)$$q_{i}=a_{k}I_{i}+b_{k},\forall i\in \omega_{k},$$
where i is the index of pixels and ω expresses the selected local square window. $a_{k}$ and $b_{k}$ are two constants in the window ω. Aiming at minimizing the reconstruction error between the input image and the output result, optimal values of $a_{k}$ and $b_{k}$ can be acquired by Equations (2) and (3).
(2)$$a_{k}=\frac{\frac{1}{\left|\omega \right|}\sum_{i\in \omega_{k}}I_{i}p_{i}-\mu_{k}{\bar{p}}_{k}}{\sigma_{k}^{2}+\varepsilon },$$
(3)$$b_{k}={\bar{p}}_{k}-a_{k}\mu_{k},$$
where $\mu_{k}$ and $\sigma_{k}$ are the mean and variance of the guided image I in the window k, respectively, and ε is a regularization parameter controlling the degree of smoothness. The values of $a_{k}$ and $b_{k}$ will determine the weight of gradient information and smoothing information. It is obvious that $b_{k}$ is approximately equal to the mean value of the pixel points when the value of a is small.

To reduce the time complexity, He et al. [20] proposed a fast guided filter on this basis. The filtering output is computed by Equation (4):(4)$${q_{i}}^{\prime }={\bar{a}}_{i}I_{i}+{\bar{b}}_{i},$$
where ${\bar{a}}_{i}$ and ${\bar{b}}_{i}$ are the averages of a and b, respectively, on the same centered window, which can be calculated by Equations (2) and (3), respectively. Firstly, the input image and the guided image are downsampled by scale s. Secondly, the output image q′ can be calculated by Equation (4). Finally, q′ is upsampled s times to obtain an output image with the same size as the original image. The time complexity of the method becomes $O(N/s^{2})$.

In this paper, a fast guided filter is applied to preprocessing before feature matching. The fast guided filter not only removes mismatch point pairs effectively, but also protects the texture information of the image preferably. The specific method is shown in Section 3.

### 2.2. Nonlinear Diffusion and the AOS Algorithm

The nonlinear diffusion method describes the luminance of an image with the increase in scale as the divergence of a flow function that controls the diffusion process [21]. This method is normally described by nonlinear partial differential equations (PDEs) owing to the nonlinear properties of the differential equations involved in diffusing the luminance of the image to the nonlinear scale space [22]. The classical nonlinear diffusion formula can be described by Equation (5):(5)$$\frac{\partial L}{\partial t}=div(c(x,y,t)\cdot \nabla L),$$
where div and ∇ represent the divergence and gradient operations, respectively.

Li et al. [23] introduced nonlinear diffusion filtering and its implementation in detail. Nonlinear diffusion is a good smoothing method, which can reduce noise while maintaining the peak shape. In order to reduce the calculation loss at the local edges, more smooth regions are selected instead of boundaries. The conduction function can be defined by Equation (6):(6)$$c(x,y,t)=g(\left.\left|\nabla L_{\sigma }(x,y,t)\right.\right|),$$
where $\nabla L_{\sigma }$ indicates the gradient of the Gaussian smoothing version of the original image. Since we chose to smooth both sides of the edge, the conduction function can be defined as follows:(7)$$g=\{\begin{array}{c} 1\\ 1-\exp(-\frac{3.315}{{\left(|\nabla L_{\sigma }|/k\right)}^{2}}) \end{array}\begin{array}{c} ,|\nabla L_{\sigma }{|}^{2}=0\\ ,|\nabla L_{\sigma }{|}^{2}>0 \end{array},$$

However, there are no analytical solutions for the PDEs involved in nonlinear diffusion filtering. The AOS algorithm uses numerical methods to approximate differential equations [24]. The diffusion equation is a semi-implicit scheme. In vector matrix representation, the discretization of Equation (5) can be expressed by Equation (8):(8)$$\frac{L^{i+1}-L^{i}}{\tau }=\sum_{i=1}^{m}A_{l}(L^{i})L^{i+1},$$
where $A_{l}$ is a matrix that encodes the image transmission of each dimension. The solution of $L^{i+1}$ can be obtained in the following way:(9)$$L^{i+1}=(I-\tau \sum_{i=1}^{m}A_{l}L^{i}{)}^{-1}L^{i},$$

The AOS algorithm is absolutely stable for any step size. Furthermore, it also creates a discrete nonlinear diffusion scale space for any large time step. The semi-implicit method greatly improves the convergence performance by reducing mask synthesis to successive one-dimensional updates represented by tridiagonal linear equations [25].

## 3. Methods

This section presents our proposed stitching method completely. Aiming at the problems of inaccurate matching and obvious splicing seams in stitched images, we propose an efficacious stitching method based on color difference and an improved KAZE with a fast guided filter. Figure 1 describes the overall process of the proposed method. Allowing for the phenomenon that feature points detected by KAZE are commonly large and intensive, we introduce a fast guide filter which uses one image as a guide map to filter the other image. This method can efficiently diminish the mismatching point pairs with preservation of original visual features. During feature matching, the KAZE algorithm based on RANSAC is utilized to match the feature points which further eliminates the mismatched points. Since the color and luminance of the images may be different, there are several clear seams in the stitching result. In order to improve the splicing effect, we produce a compensation method based on color and brightness to calculate the color difference in the overlapping area. Then, the obtained local color and luminance compensation values are applied to the overall image to be spliced. In addition, our proposed method can realize the stitching of multiple images. In order to avoid excessive variation, we use the middle image as the reference image to warp the images.

### 3.1. Fast Guided Filtering

The first image is imported as a guide map to repair the texture information in the other image. The purpose of introducing fast guided filtering is to decline the mismatch rate of the KAZE algorithm. Here, we select a window size of 8 pixels and set the regularization parameter that controls the smoothness level to 0.01. The sampling ratio is set to four pixels in the fast guided filter. Figure 2 shows the results of fast guided filtering.

Figure 2a,b contains input images, where (a) is the original image and (b) is the guide image. The fast guided filter filters redundant information by analyzing common textures and spatial features in both the original image and guide image. Additionally, then the fast guided filter can restore the boundaries while smoothing the original image by the guide map. This method greatly reduces the number of feature matching pairs, which results in a lower mismatch rate. After guided filtering, a lot of invalid information in the image can be filtered out, which widely improves the iterative efficiency during matching. At the same time, it can also reduce the mismatch rate to a certain extent.

### 3.2. KAZE Algorithm Based on RANSAC

Both the SIFT algorithm and the SURF algorithm detect feature points in a linear scale space, which can notably cause blurred boundaries and loss of detail. The KAZE algorithm detects feature points in the nonlinear scale space by constructing the nonlinear scale space, which retains more image detail. In the KAZE algorithm, AOS technology and variable transfer diffusion are used to establish a nonlinear scale space for the input images. Then, the 2D features of interest that exhibit the maximum value of scale-normalized determinant detection of Hessian response are detected by the nonlinear scale space. Finally, the scale and rotation invariance scriptors considering the first-order image derivatives are obtained through calculating the principal direction of the key point.

We introduce the RANSAC algorithm on the basis of the traditional KAZE algorithm, which can further eliminate mismatched point pairs. The flow of the KAZE algorithm based on RANSAC is shown in Figure 3.

#### 3.2.1. Construction of Nonlinear Scale Space

The KAZE algorithm constructs a nonlinear scale space through nonlinear diffusion filtering and the AOS algorithm [26]. The scale of KAZE features increases logarithmically. Each level of the KAZE algorithm adopts the same resolution as the original image. According to the principle of the difference of Gaussian (DoG) pyramid model [27], each group needs S+2 layers of images to detect the extreme points of S scales. Then, the parameter of scale space, θ, can be described as follows:(10)$$\theta (o,s)=\theta_{0}2^{o+\frac{s}{S}},o\in [0,1,\ldots ,O-1],s\in [0,1,\ldots ,S+2],$$
where $\theta_{0}$ is the baseline scale, o is the index of the group octave and S represents the index of the intra-group layer. The scale parameters of key points are calculated according to the group of key points and the number of layers in the group in combination with Equation (10).

The scale parameters of each layer in the group for constructing the Gaussian pyramid are calculated according to the following formulas:(11)$$\theta (s)=\sqrt{{\left(r^{s}\theta_{0}\right)}^{2}-{\left(r^{s-1}\theta_{0}\right)}^{2}},$$
(12)$$r=2^{\frac{1}{S}},$$

From the above formulas, the scale parameters of the same layer in different groups are the same. The scale calculation formula of a layer of images in the group is shown in Equation (13).
(13)$$\theta (s)=\theta_{0}2^{\frac{s}{S}},s\in [0,1,\ldots ,S+2],$$

To ensure the continuity of the scale space, the first image of a group is obtained by sampling the penultimate layer of the previous group. We assume that the initial scale of the first group is θ. Then, the scale parameters of each layer in the first group are θ, rθ, $r^{2}\theta$ and so on. The scale of the penultimate layer can be defined as follows:(14)$$r^{s}\theta =2\theta ,$$

Since the nonlinear diffusion filter model is based on time, it is necessary to convert the scale parameters into evolution time. We suppose that the standard deviation used in Gaussian scale space is σ. Then, the convolution of the Gaussian is equivalent to filtering the image with a duration of $t=\sigma^{2}/2$. According to a set of evolution time, all images in the nonlinear scale space can be obtained by the AOS algorithm as follows:(15)$$L^{i+1}=[I-(t_{i+1}-t_{i})\sum_{i=1}^{m}A_{i}(L^{i}){]}^{-1}L^{i},$$

#### 3.2.2. Detection and Location of Feature Points

The feature points are obtained by seeking the local maximum points of the Hessian determinant normalized by different scales. For multi-scale feature detection, we normalize the set of differential operators for scale:(16)$$L_{Hessian}=\theta^{2}(L_{xx}L_{yy}-L_{xy}^{2}),$$
where $L_{xx}$ and $L_{yy}$ are the second-order horizontal derivative and vertical derivative, respectively, and $L_{xy}$ expresses the second-order cross derivative. For the filtered image set in a nonlinear scale space, we analyze the response of the detector at different scale levels $\theta_{i}$. The maximum values of scale and spatial position are searched in all filtered images except i=0 and i=N. Then, we check the response in a 3×3 pixels window to quickly find the maximum value.

As shown in Figure 4, each feature point is compared with 26 pixel points from the same layer and two adjacent layers. All pixels are traversed in this way until the maximum point is found. The derivative is calculated by the Taylor expression.
(17)$$L(x)=L+(\frac{\partial L}{\partial x}{)}^{T}x+\frac{x^{T}}{2}(\frac{\partial^{2}L}{\partial^{2}x})x,$$

We take the derivative of the above equation and make it equal to zero. Then, the solution can be obtained by the following.
(18)$$\hat{x}=(\frac{\partial^{2}L}{\partial x^{2}}{)}^{-1}\frac{\partial L}{\partial x},$$
(19)$$L(\hat{X})=L+\frac{1}{2}(\frac{\partial L}{\partial x}{)}^{T}\hat{X},$$

One of the feature points that meets $L(\hat{X})\geq T$ is selected as the key point.

#### 3.2.3. Determination of Main Direction

Assuming that the scale parameter of the characteristic point is $\theta_{i}$, the search radius is set to $6\theta_{i}$. We make six 60 degree sector regions in this radius and count the sum of Haar wavelet features in each sector region. The direction with the largest sum of wavelet features is the main direction.

#### 3.2.4. Generation of Feature Descriptors

For feature points whose scale parameter is $\theta_{i}$, we take a window of $24\theta_{i}\times 24\theta_{i}$ in the gradient image centered on the feature point. Then, the window is divided into 4 × 4 sub-regions with a size of $9\theta_{i}\times 9\theta_{i}$. Adjacent sub-areas have an overlapping zone with a width of $2\theta_{i}$. In the process of calculating the sub-region description vector, a Gaussian kernel with a weight of $\theta_{1}=2.5\theta_{i}$ is used for each sub-region to obtain the description vector as Equation (20):(20)$$d_{v}=(\sum L_{x},\sum L_{y},\sum \left|L_{x}\right|,\sum \left|L_{y}\right|),$$

After that, the vector $d_{v}$ of each sub-region is weighted by another 4 × 4 Gaussian window where $\theta_{2}=1.5\theta_{i}$. Finally, a 64-dimensional description vector is attained through normalization processing.

#### 3.2.5. Eliminate Mismatched Point Pairs

In the case of information loss caused by image blur, noise interference and compression reconstruction, the robustness of KAZE feature point detection is significantly better than other features. In addition, the nonlinear scale space does not cause boundary blurring and detail loss compared to the linear scale space. However, the matching of KAZE features is sensitive to the setting of parameters, which can easily prompt mismatching. Aiming at the problem of excessive concentration and mismatch of feature points, the RANSAC method is introduced to effectively eliminate mismatched points. Firstly, KAZE feature matching is performed on the previous input image and the filtered image. Additionally, then RANSAC algorithm is applied to eliminate the mismatched point pairs.

The RANSAC algorithm is used to find an optimal homography matrix. The standardized matrix satisfies Equation (21).
(21)$$s\left[\begin{array}{c} x^{\prime }\\ y^{\prime }\\ 1 \end{array}\right]=\left(\begin{array}{ccc} h_{11} & h_{12} & h_{13}\\ h_{21} & h_{22} & h_{23}\\ h_{31} & h_{32} & h_{33} \end{array}\right)\left[\begin{array}{c} x\\ y\\ 1 \end{array}\right],$$

To calculate the homography matrix, we randomly extract four samples from the matching dataset which are not collinear. Then, the model is utilized for all datasets to calculate the number of points and the projection error. If the model is optimal, the loss function is the smallest. The loss function can be calculated by Equation (22).
(22)$$\sum_{i=1}^{n}x_{i}^{\prime }\frac{h_{11}x_{i}+h_{12}y_{i}+h_{13}}{h_{31}x_{i}+h_{32}y_{i}+h_{33}}+(y_{i}^{\prime }\frac{h_{21}x_{i}+h_{22}y_{i}+h_{23}}{h_{31}x_{i}+h_{32}y_{i}+h_{33}}{)}^{2},$$

The process of the RANSAC algorithm is shown in Algorithm 1.
**Algorithm 1:** The process of RANSACInput: The feature point pair S of the image to be spliced and the maximum iteration number k.Output: Remove the feature point pair D of mismatched point pairs.1: S->D.2: Repeat:3: Randomly extract four sample data from all feature point pairs detected, which cannot be collinear.4: Use Equations (21) and (22) to calculate the transformation matrix H, and record it as model M.5: Calculate the projection error between all data in the dataset and model M. If the error is less than the threshold, add the interior point set I.6: If the number of elements of the current interior point set I is greater than the optimal interior point set I_ Best, update I_ Best = I, update the number of iterations at the same time k=log(1−p)log(1−ωm).7: Until iterations are greater than the iteration number k.

The iteration number k is constantly updated rather than fixed when it is not greater than the maximum iteration number. The formula for calculation of the iteration number is as follows:(23)$$k=\frac{\log(1-p)}{\log(1-\omega^{m})},$$
where p expresses the confidence level, ω is the proportion of the interior point and m is the minimum number of samples required to calculate the model.

The RANSAC algorithm performs precise matching of feature points through the above iterative process, which effectively eliminates mismatched point pairs. The inner point pairs obtained by the method are the most advantageous pairs. The matching accuracy is the foundation for accurately evaluating warped equations. If there are many mismatched pairs, it is easy for problems such as ghosting and distortion to occur in the splicing result.

### 3.3. Color Difference Compensation

The luminance and color of the image to be stitched may be uneven, which can easily result in obvious stitching seams in the stitching result. To improve this problem, we put forward a color difference compensation method. The process is shown in Figure 5.

We calculate the color difference and luminance difference of the overlapping area of the two distorted images. Here, we chose the LAB color model. The average color difference is calculated as follows:(24)$$L_{avg}=\frac{1}{n-1}\sum_{i=1}^{n}(L_{i1}-L_{i2}),$$
(25)$$A_{avg}=\frac{1}{n-1}\sum_{i=1}^{n}(A_{i1}-A_{i2}),$$
(26)$$B_{avg}=\frac{1}{n-1}\sum_{i=1}^{n}(B_{i1}-B_{i2}),$$
where $L_{avg}$ represents the average brightness difference, $L_{i1}$ is the brightness of the overlapping area of the first image and $L_{i2}$ is the brightness of the overlapping area of the second image. Similarly, $A_{avg}$ and $B_{avg}$ are the average color difference, $A_{i1}$ and $B_{i1}$ are the colors of the overlapping area of the first image and $A_{i2}$ and $B_{i2}$ are the colors of the overlapping area of the second image. We chose the image with the higher brightness as the standard to compensate for the global brightness and color of another image. Finally, the compensated distorted image is fused to obtain the final mosaic image. Figure 6 shows the original images and Figure 7 shows the comparison of color compensated and uncompensated splicing results.

It can be seen from the figure that the stitching effect after color compensation is smoother. Not only does the stitching result have no obvious seams, but also has an improved uniformity.

## 4. Experiments

This section introduces some experiments to evaluate our proposed splicing method. The images used in the experiment are mainly from the ground truth database of the University of Washington [28] and the USI-SIPI image database of the University of Southern California [29]. The ground truth database of the University of Washington addresses the need for experimental data to quantitatively evaluate emerging algorithms. The high-quality and high-resolution color images in the database represent valuable extended duration digitized footage to those interested in driving scenarios or ego-motion. The images in the USI-SIPI image database of the University of Southern California have been provided for research purposes. The USI-SIPI image database contains multiple types of image sets, including texture images, aerial images, miscellaneous images and sequence images. Most of the material was scanned many years ago in the research group from a variety of sources.

At the same time, all test methods are run on a 2.60 GHz CPU with 16 GB RAM under the same experimental settings in this paper. In order to better evaluate the algorithm, we compared subjective images and quantitative evaluation. We compare the results with several popular splicing methods.

### 4.1. Intuitive Effect

We took a group of images in the USI-SIPI image database of the University of Southern California as an example to compare our method with SURF, BRISK, Harris, MinEigen, MSER and ORB matching algorithms. The splicing effect is shown in Table 1.

The upper right corner of the stitched image in the table shows a magnification of the area marked in blue. It is not difficult to see from the table that the proposed algorithm is not only better than other algorithms regarding detail, but also more uniform in the color and brightness of the mosaic image. As shown in the red marked area in the table, the stitching method we proposed has no obvious stitching seams, while other algorithms have several obvious stitching seams.

In addition, our method can be applied to multiple image stitching. We selected multiple UAV images from the image library, and the splicing effect is shown in Figure 8.

### 4.2. Quantitative Evaluation

In order to effectively evaluate the effectiveness of the algorithm, we quantitatively evaluated the above algorithms from four aspects: the number of matching pairs, the correct matching rate, the root mean square error (RMSE) and the mean absolute error (MAE) [30,31,32].

Assuming that the total number of matches is N and the correct matching logarithm is A, the correct matching rate can be defined as Equation (27).
(27)$$Accurancy=\frac{A}{N}\times 100\%,$$

We randomly selected eight groups (Table 2) of data from the two databases for feature matching. The table shows the matching logarithms and the matching accuracy results of several comparison algorithms.

It can be seen from the above table that the extent of feature matching based on ORB is the largest. Although the overall matching of the ORB is relatively stable, the matching rate is lower than the proposed method in this paper. Other comparison algorithms detect fewer feature points. Even though other algorithms have high matching rates for individual data, their performance in terms of the number of feature points, matching accuracy and stability is poor. The improved KAZE algorithm based on a fast guided filter is second only to the ORB algorithm in the number of feature matching pairs. Our proposed method further filters out the mismatched point pairs, thereby improving the matching efficiency and matching stability.

In order to judge the stability of the matching algorithm more intuitively, we compared the accuracy of several algorithms in a line chart (Figure 9).

The red color in the figure shows the matching accuracy of our proposed algorithm. It is not difficult to see from the figure that the algorithm we proposed is not only more accurate, but also more stable.

For two images, I_1_ and I_2_, to be spliced, given N matching pairs (pi1 and pi2), where i = 1, 2, N, the RMSE and the MAE are defined as follows:(28)$$RMSE(I,I^{\prime })=\sqrt{\frac{1}{N}\sum_{i=1}^{N}\left(f(p_{i}\right)-p_{i}^{\prime }{)}^{2}},$$
(29)$$MAE(I,I^{\prime })=\frac{1}{N}\sum_{i=1}^{N}\left|f(p_{i})-p_{i}^{\prime }\right|,$$

Additionally, taking the selected eight groups (Table 3) of data as an example for splicing, the table shows the comparison results of the RMSE and the MAE of several different algorithms.

The table shows that the RMSE and the MAE values of our proposed method are lower. The overall stitching effect is better than other stitching algorithms.

In addition, we also compared the processing time of the proposed algorithm with other algorithms. The result shows that the processing time of the proposed algorithm is similar to that of other algorithms. The comparison results of processing time are shown in Table 4.

It is not difficult to draw the conclusion from Table 4 that the processing time of the SURF algorithm is relatively low and the ORB algorithm is the most time consuming among the studied algorithms. The proposed method has similar processing times compared to other methods.

## 5. Conclusions

In this paper, we propose an improved image stitching algorithm based on color difference and KAZE with a fast guided filter, which solves the problems of a high mismatch rate in feature matching and obvious seams in the stitched image. In this paper, a fast guide filter is introduced to reduce the mismatch rate before matching. The KAZE algorithm based on RANSAC is used for feature matching of the image to be spliced, and the matrix transformation of the image to be spliced is performed. Then, the color difference and brightness difference of the overlapping area of the image to be spliced is calculated, and overall adjustments of the image to be spliced are made so as to improve the nonuniformity of the spliced image. Finally, the converted image is fused to obtain the stitched image.

Our proposed method was evaluated via the resulting visual image and quantitative value of the images of the ground truth database and USI-SIPI image database and compared with other popular algorithms. On the one hand, the stitching method we proposed can achieve smoother and more detailed stitched images. On the other hand, the algorithm proposed in this paper is superior to other algorithms in terms of matching accuracy, the RMSE and the MAE.

The proposed stitching method in this paper can be used in many fields, such as UAV panoramic stitching and virtual reality. However, the limitations of our proposed method are as follows: firstly, the method proposed in this paper is prone to ghosting when splicing moving objects in close proximity. Secondly, the proposed algorithm has a similar processing time compared to other algorithms, but cannot achieve the effect of real-time stitching. In the future, we will further improve the efficiency of image stitching.

## Figures and Tables

**Figure 1 sensors-23-04583-f001:**
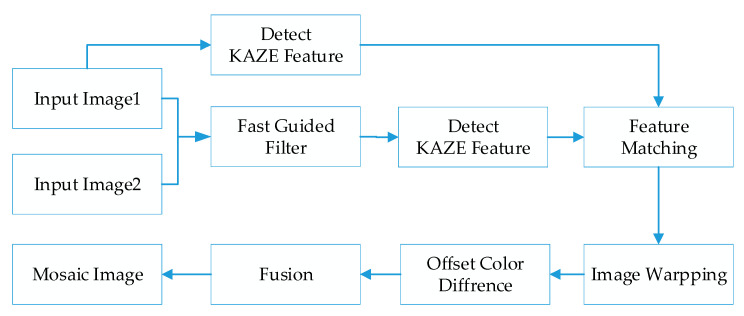
The proposed framework.

**Figure 2 sensors-23-04583-f002:**
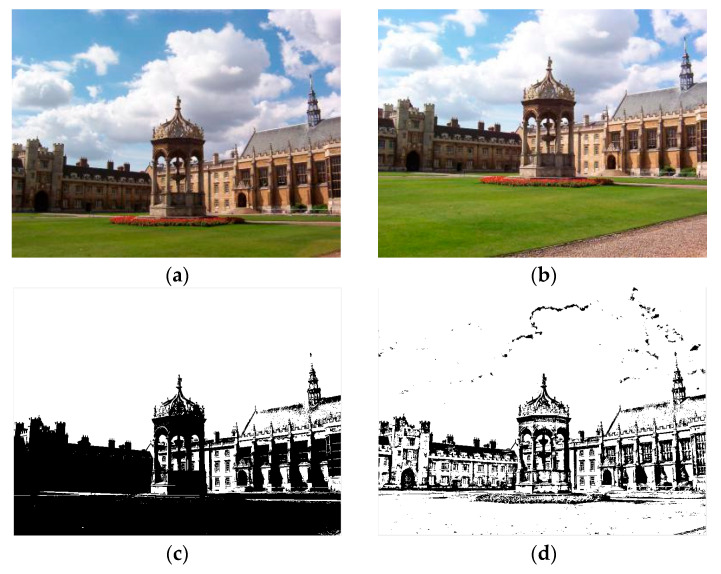
(**a**) is the original image; (**b**) is the guide image; (**c**) is the binary image without filtering; (**d**) is the filtered binary image.

**Figure 3 sensors-23-04583-f003:**
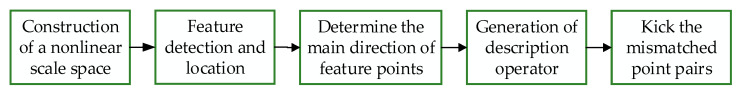
The flow of KAZE algorithm based on RANSAC.

**Figure 4 sensors-23-04583-f004:**
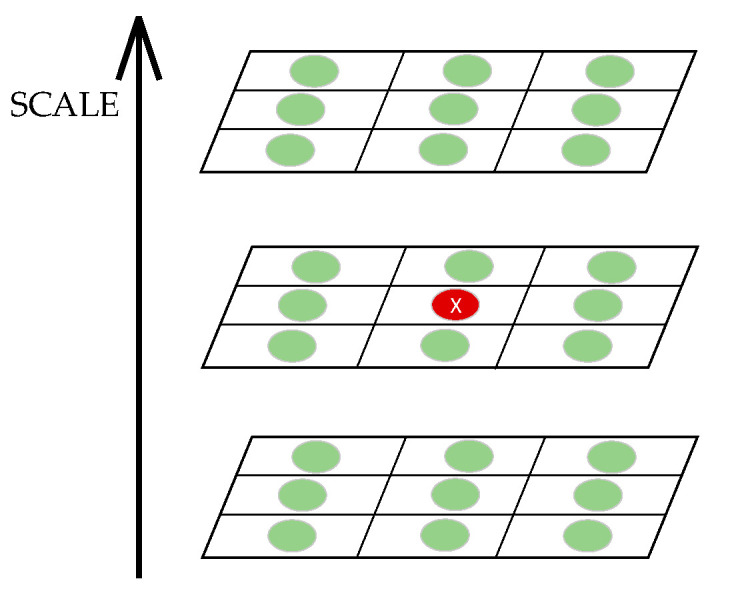
Detection and location of each feature point.

**Figure 5 sensors-23-04583-f005:**
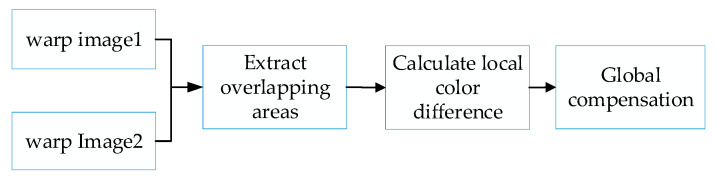
The flow of the color difference compensation.

**Figure 6 sensors-23-04583-f006:**
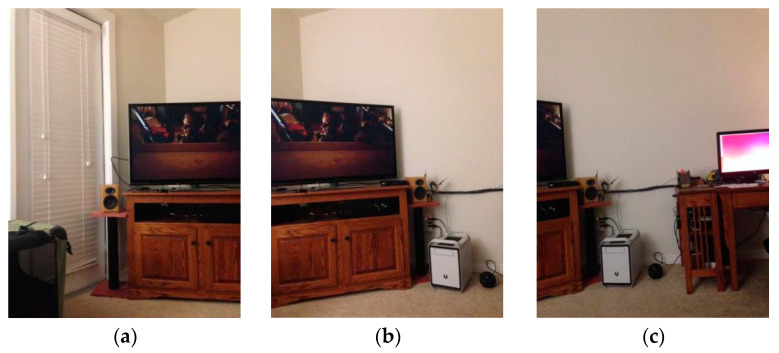
Original images. (**a**–**c**) are the original images.

**Figure 7 sensors-23-04583-f007:**
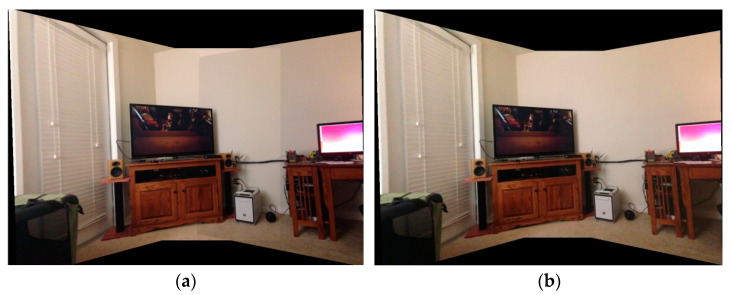
(**a**) Mosaic image without color compensation. (**b**) Mosaic image with color compensation.

**Figure 8 sensors-23-04583-f008:**
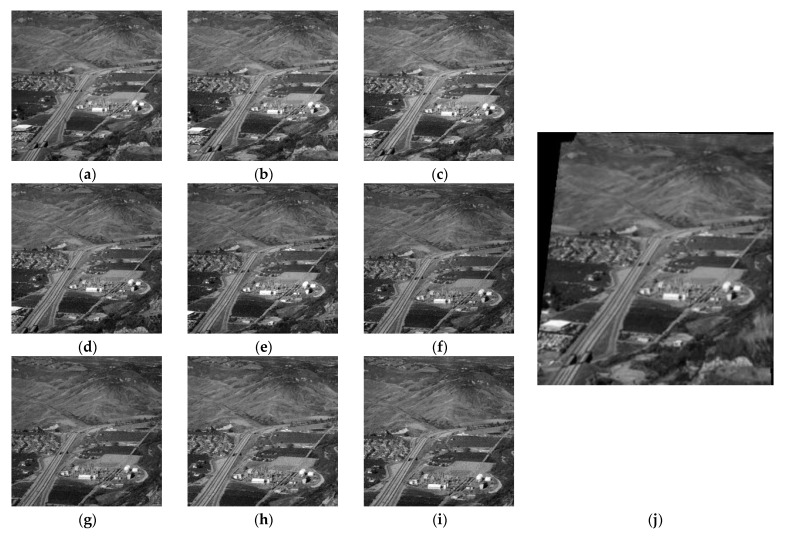
UAV images panoramic mosaic: (**a**–**i**) are the original images; (**j**) is the stitched image.

**Figure 9 sensors-23-04583-f009:**
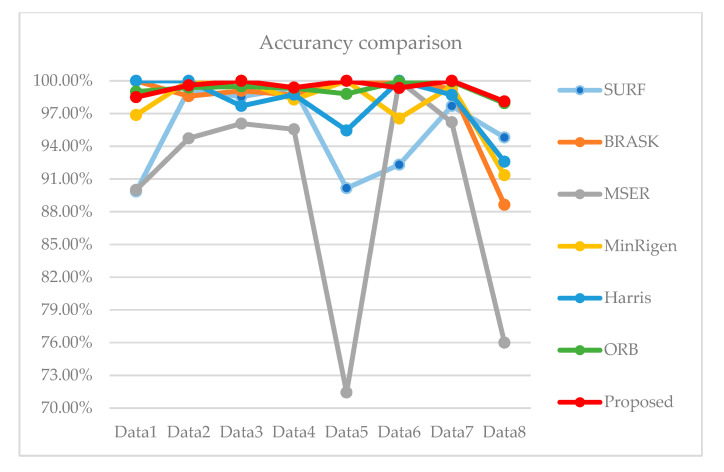
Comparison of accuracy.

**Table 1 sensors-23-04583-t001:** Splicing effect.

Methods	Match Figure	Stitched Image
SURF [9]	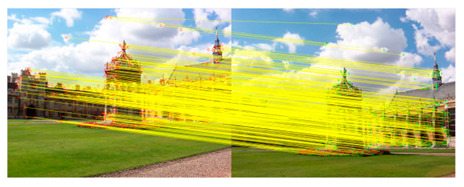	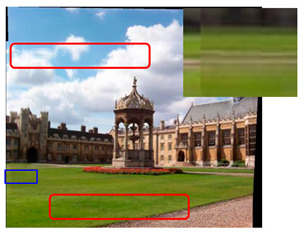
BRISK [11]	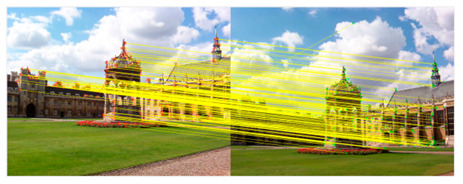	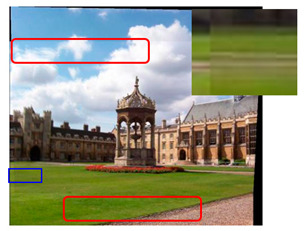
Harris [14]	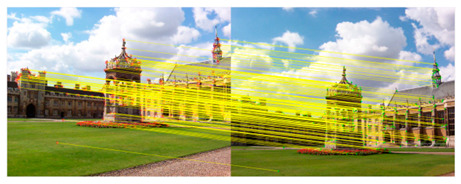	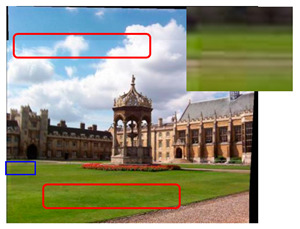
MinEigen	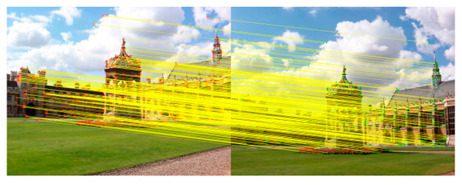	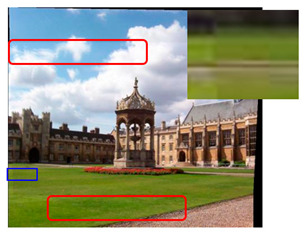
MSER [13]	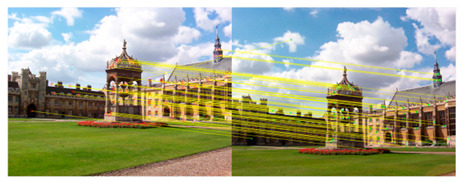	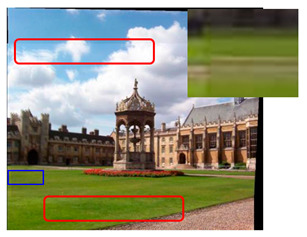
ORB [12]	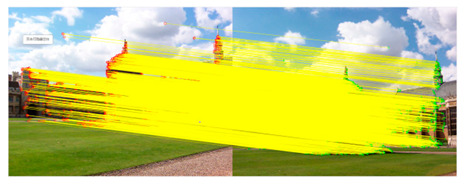	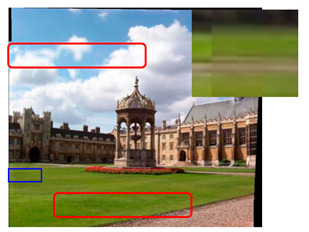
Our method	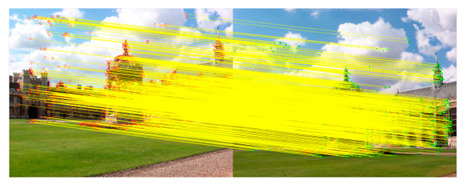	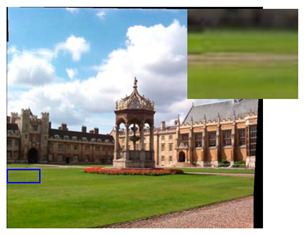

**Table 2 sensors-23-04583-t002:** The results of a comparison of different algorithms.

Data	Estimate	SURF	BRASK	MSER	MinEigen	Harris	ORB	Proposed
Data1	Matching pairs	69	14	20	32	19	101	267
	Accuracy	89.85%	**100%**	90.00%	96.87%	**100%**	99.01%	98.50%
Data2	Matching pairs	123	71	19	68	36	321	243
	Accuracy	99.18%	98.59%	94.73%8	**100%**	**100%**	99.38%	99.59%
Data3	Matching pairs	205	109	51	125	87	771	606
	Accuracy	98.54%	99.08%	96.08%	100%	97.70%	99.48%	**100%**
Data4	Matching pairs	474	231	135	232	157	2348	1280
	Accuracy	99.36%	98.70%	95.56%	98.28%	98.73%	99.32%	**99.38%**
Data5	Matching pairs	61	38	7	39	22	167	148
	Accuracy	90.16%	100%	71.43%	100%	95.45%	98.80%	**100%**
Data6	Matching pairs	39	15	10	29	19	133	154
	Accuracy	92.31%	**100%**	**100%**	96.55%	**100%**	99.93%	99.35%
Data7	Matching pairs	564	266	79	398	233	3112	1242
	Accuracy	97.69%	99.25%	96.20%	99.50%	98.71%	99.94%	**100%**
Data8	Matching pairs	154	44	25	104	54	440	212
	Accuracy	94.81%	88.64%	76.00%	91.35%	92.59%	97.95%	**98.11%**

**Table 3 sensors-23-04583-t003:** The comparison results of the RMSE and the MAE of different algorithms.

Data	Estimate	SURF	BRASK	MSER	MinEigen	Harris	ORB	Proposed
Data1	RMSE	**0.1037**	0.1354	0.1301	0.1321	0.1309	0.1261	0.1237
	MAE	**0.0759**	0.1157	0.1051	0.1060	0.1096	0.0935	0.0935
Data2	RMSE	0.0917	0.1367	0.0973	0.1173	0.1185	0.1099	**0.0797**
	MAE	0.0671	0.0933	0.0819	0.0916	0.0846	0.0833	**0.0649**
Data3	RMSE	0.1285	0.1439	0.1386	0.1738	0.1727	0.1831	**0.1283**
	MAE	**0.0992**	0.1155	0.1243	0.1370	0.1387	0.1505	0.0999
Data4	RMSE	0.0920	0.0946	0.0966	0.1616	0.1734	0.1417	**0.0903**
	MAE	0.0592	0.0655	0.0597	0.1124	0.1194	0.0905	**0.0545**
Data5	RMSE	0.0776	0.0895	0.2573	0.0867	0.8250	0.1169	**0.0760**
	MAE	0.0588	0.0650	0.1305	0.0708	0.0717	0.0769	**0.0586**
Data6	RMSE	0.0539	0.0713	0.0562	0.0599	0.0482	0.0691	**0.0417**
	MAE	0.0398	0.0510	0.0467	0.0429	0.0394	0.0458	**0.0302**
Data7	RMSE	0.1074	0.1120	0.1386	0.1392	0.1222	0.1494	**0.1038**
	MAE	0.0847	0.0888	0.1084	0.1040	0.0947	0.1075	**0.0842**
Data8	RMSE	0.0705	0.0719	0.0400	0.1139	0.1047	0.0994	**0.0701**
	MAE	0.0494	0.0496	0.0345	0.0777	0.0812	0.0666	**0.0447**

**Table 4 sensors-23-04583-t004:** The comparison results of processing time of different algorithms.

Data	SURF	BRASK	MSER	MinEigen	Harris	ORB	Proposed
Data1	**1.226 s**	1.681 s	1.456 s	1.772 s	1.354 s	1.879 s	1.798 s
Data2	2.239 s	3.001 s	2.489 s	2.096 s	**2.204 s**	5.370 s	3.232 s
Data3	1.168 s	**1.444 s**	1.549 s	2.238 s	1.179 s	2.523 s	2.389 s
Data4	**1.219 s**	1.341 s	1.318 s	1.996 s	1.487 s	2.653 s	1.979 s
Data5	**1.149 s**	1.362 s	1.251 s	1.359 s	1.160 s	2.568 s	1.356 s
Data6	1.715 s	2.406 s	1.998 s	1.337 s	**1.181 s**	1.661 s	1.509 s
Data7	**1.089 s**	1.964 s	1.981 s	1.549 s	1.347 s	2.017 s	1.973 s
Data8	**2.147 s**	3.010 s	2.984 s	2.471 s	2.309 s	4.175 s	2.992 s

## Data Availability

Not applicable.

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
