# Peer review of "Image Stitching Based on Color Difference and KAZE with a Fast Guided Filter"

_sensors, 2023, doi:10.3390/s23104583_

Round 1

Reviewer 1 Report

The authors proposed an image stitching algorithm based on color difference and KAZE enhanced with fast guided filtering.

-        Page 2 – line 54. OrdoñezA is not correctly cited.

-        Page 5 – line 183. in KAZE, i must be in capital letters.

-        Page 9 – line 311. Figure 6. original images. After the period, the first letter must be in capital letters.

-        Page 9 – line 312. In figure 7a and 7b, specifically in figure caption, both are without color compensation?

-        In the first paragraph of experiments, describe in general terms the two databases used.

-        Page 13 – Figure 9. Please make the figure in a specialized software.

-        Discussion/Conclusion – What are the limitations of the proposed method? Please write them.

-        Speaking of computational time, what results does it present with respect to the other methods. Make a comparative table.

Author Response

Dear reviewer,

Thank you for your pertinent comments. Based on your comments, we have made the following modifications:

  • I'm sorry for the citation error caused by personal carelessness. We have rechecked the cited reference and changed OrdoñezAto OrdoñezOn page 2, line 60.
  • On page 6, line 208, we modifythe ‘i’in 'in KAZE' to capital letters.
  • In Figure 6. We modify the first letter of ‘original images’to capital letters.
  • In the description of Figure 7bin page 10,We change the original to 'with color compensation'.
  • In the first paragraph of experiments in page 10, we briefly describe the Ground truth database of the University of Washington and the USI-SIPI image database of the University of Southern California. We simply introduce these two databases in terms of source, target, and image type.
  • We have tried to usespecialized software such as MATLAB to create the But we find that the effect of using Excel is clearer and easier to understand. Therefore, we still use the figure made in Excel and make minor adjustments on the original basis.
  • Based on your suggestion, we supplementthe limitations of the proposed method in the conclusion section. The limitations of our proposed method are as follows: firstly, the method proposed in this paper is prone to ghosting when splicing moving objects in close proximity. Secondly, the proposedalgorithm has a similar processing time compared to other algorithms, but cannot achieve the effect of real-time stitching.
  • In addition, we add comparative experiments on the processing time with other algorithmsin the conclusion section. We still use the eight sets of data mentioned in the previous article as examples to compare the processing time of several algorithms. The specific results are shown in Table 5.

Thank you again for your valuable comments and look forward to your reply.

Kind regards,

Chong Zhang

Reviewer 2 Report

Dear authors.,

Overall, your abstract is well-written and provides a clear overview of your proposed image-stitching algorithm. You have effectively highlighted the importance of image stitching and stated the purpose of your research. You have also mentioned the critical components of your algorithm, such as the use of fast guided filter and KAZE algorithm, and how they contribute to improving the stitching effect. You have also described how your algorithm addresses the issues of mismatch rate and nonuniformity in the splicing result, which is important for image-stitching applications. It is good that you evaluated your algorithm using visual effect maps and quantitative values, which provides a comprehensive analysis of the algorithm's performance. However, it would be helpful to provide more details on the specific improvements made to the KAZE algorithm based on improved Random Sample Consensus, as this could be a key factor in the success of your algorithm. Additionally, it would be beneficial to mention the dataset used for evaluation and any limitations of your proposed algorithm that may need to be addressed in future research. Overall, your abstract is well-written and provides a clear overview of your proposed algorithm and its performance.

Firstly, it must explain the motivation behind the research and why image stitching is an important problem to solve. the author can highlight the potential applications of image stitching, such as in augmented reality, ground reconnaissance, and moving object detection and tracking.

Next, the author can provide more details on the specific components of your proposed algorithm, such as the fast-guided filter and KAZE algorithm with improved Random Sample Consensus. it can explain how these components work and why they are effective in improving the stitching effect and reducing the mismatch rate too.

the authors can also provide more information on the evaluation of your algorithm, including the specific metrics used for quantitative evaluation and any potential limitations or biases of the dataset used.

Finally, the author can summarize the key findings and contributions of your research, highlighting how your proposed algorithm improves upon existing approaches and its potential impact in real-world applications.

Author Response

Dear reviewer,

Thank you for your pertinent comments. Based on your comments, we have made the following modifications:

Firstly, we add the application areas and importance of image stitching in lines 28-35 of the introduction. We discuss the applications and the importance of the image stitching from the following aspects. On the one hand, image stitching can synthesize panorama and ultra wide view images, so that ordinary cameras can achieve grand scenes shooting. On the other hand, image stitching can synthesize fragmented images into a complete image. The stitching technology can be applied to combine medical images, scientific microscope fragments, or local images from seabed exploration into a complete image. In addition, image mosaic is also a basic technology in scene rendering methods, which uses panoramic images instead of 3D scenes to model and draw.

Secondly, we explain the reason of the fast guided filter can reduce the mismatch rate from line 195 to 203 of section 3.1. The fast guided filter filters redundant information by analyzing common texture and spatial features in both the original image and guide image. And then the fast guided filter can restore the boundaries while smoothing the original image by the guide map. This method greatly reduces the number of feature matching pairs, which resulting in a lower mismatch rate. After guided filtering, a lot of invalid information in the image can be filtered out which widely improve the iterative efficiency during matching. In addition, the principle of the KAZE algorithm with improved Random Sample Consensus algorithm has been detailed in Section 3.2. Based on your suggestion, we have added the reasons and importance of KAZE algorithm with improved RANSAC from line 316 to 321 of section 3.2.5, which can reduce the mismatch rate and improve the splicing effect. The RANSAC algorithm performs precise matching on feature points through the above iterative process, which effectively eliminating mismatched point pairs. Matching accuracy is the foundation for accurately evaluating warped equations. If there are many mismatched pairs, it is easy to cause problems such as ghosting and distortion in the splicing result.

Finally, We supplement the problems solved by the proposed algorithm in the conclusion. We emphasize the improvements and contributions of the proposed algorithm. In addition, we have also improved the description of the limitations and potential impacts of the proposed algorithm. Due to the inability to achieve real-time splicing mentioned in this section, a quantitative evaluation of processing time has been added in this section. We still use the eight sets of data mentioned in the previous article as examples to compare the processing time of several algorithms. The specific results are shown in Table 5.

Thank you again for your valuable comments and look forward to your reply.

Kind regards,

Chong Zhang

Reviewer 3 Report

In this paper, the authors present an image stitching algorithm based on color difference and an improved KAZE feature-matching method with a fast-guided filter. The proposed algorithm first employs a fast-guided filter to reduce the mismatch rate before feature matching, next, it utilizes an improved KAZE algorithm based on Random Sample Consensus for feature matching. Finally, the warped images with colour difference compensation are fused to produce the stitched image. The proposed method is evaluated using visual effect maps and quantitative values and compared to other popular stitching algorithms. Despite the good use of the English language and a good structure of the paper, many concepts need to be focused on improving paper quality:

-       In the introductory section, the contributions, limitations and novelty of this work are not clear or little specified. I suggest you try to include more bibliographical references for dealing with this topic to make a proper comparison and under light your contributions to the research;

-       From a technical standpoint, the literature reveals that machine-learning methods have been successfully incorporated into image-stitching techniques. This integration significantly enhances the overall quality and accuracy of the resulting stitched images, making these approaches highly suitable for a wide range of applications. Please to well motivate the missing of use of ML techniques in this work.

-       In the conclusion section, I suggest supporting the achieved results with appropriate numerical results and references, offering a more critical/discursive view of future research;

Author Response

Dear reviewer,

Thank you for your pertinent comments. Based on your comments, we have made the following modifications:

Firstly, We complement the importance and application of image stitching in the first paragraph of the introduction. On the one hand, image stitching can synthesize panorama and ultra wide view images, so that ordinary cameras can achieve grand scenes shooting. On the other hand, image stitching can synthesize fragmented images into a complete image. The stitching technology can be applied to combine medical images, scientific microscope fragments, or local images from seabed exploration into a complete image. In addition, image mosaic is also a basic technology in scene rendering methods, which uses panoramic images instead of 3D scenes to model and draw.

Secondly, the contribution of the proposed algorithm is detailed at the end of the introduction in this article, so we apologize that the description of the contribution section in this article has not been adjusted.

Thirdly, We supplement the problems solved by the proposed algorithm in the conclusion.  We have also improved the description of the limitations and potential impacts of the proposed algorithm. Due to the inability to achieve real-time splicing mentioned in this section, a quantitative evaluation of processing time has been added in this section. We still use the eight sets of data mentioned in the previous article as examples to compare the processing time of several algorithms. The specific results are shown in Table 5. Therefore, the limitations of the proposed algorithm were not supplemented in the introduction of this article, otherwise it may lead to duplication.

In addition, we add the descriptions and references to image stitching algorithms based on deep learning in lines 69 to 77 of the introduction. We include references to currently effective image stitching algorithms based on deep learning and list the advantages. At the same time, we analyzed the shortcomings of image stitching algorithms based on deep learning. The stitching algorithms based on machine learning require extensive training, high resource consumption, and long time consumption. Therefore, image stitching based on deep learning is not used in this paper.

Finally, We supplement the problems solved by the proposed algorithm in the conclusion section. We emphasize the improvements and contributions of the proposed algorithm. In addition, we also improve the description of the limitations and potential impacts of the proposed algorithm in this paper. Due to the inability to achieve real-time splicing mentioned in this section, a quantitative evaluation of processing time has been added in this section. Based on your suggestion, we still use the eight sets of data mentioned in the previous article as examples to compare the processing time of several algorithms. The specific results are shown in Table 5.

Thank you again for your valuable comments and look forward to your reply.

Kind regards,

Chong Zhang

Round 2

Reviewer 1 Report

The comments have been satisfactorily addressed. However, the time comparison table must be in the discussion section. Please change it.

Author Response

Dear expert,

  Thanks for your valuable comment. Based on your comment, we have changed the time comparison table to the discussion section of the experiment.

  Thank you again for your valuable feedback. Looking forward to your reply.

Kind regards,

Chong Zhang

Reviewer 3 Report

The article proposes an image stitching algorithm based on colour difference and an improved KAZE algorithm with a fast-guided filter. The proposed method is evaluated both visually and quantitatively and compared with other popular stitching algorithms.  This improved version of the manuscript presents improved results and a good level of English. Moreover, the authors added and followed all my last suggestions parts. All essential aspects for reading and reproducibility of the work have been added. However, the statement highlights that a direct comparison with state-of-the-art methods has not been made, so it would be beneficial to compare the proposed algorithm with more advanced techniques.

Author Response

Dear expert,

  Thanks for your valuable comment. The comparative experiments in the paper are all effective algorithms proposed in recent years. However, the method proposed in this paper has not been compared with algorithms based on deep learning, due to the excessive resource and time consumption of stitching algorithms based on deep learning. In addition, the evaluation criteria for deep learning based stitching algorithms are also different from the RMSE, MAE mentioned in this paper. In the future, we will delve deeper into image stitching algorithms and compare them with algorithms based on deep learning.

Thank you again for your valuable feedback. Looking forward to your reply.

Kind regards,

Chong Zhang